# The Mechanisms of Parental Burnout Affecting Adolescents’ Problem Behavior

**DOI:** 10.3390/ijerph192215139

**Published:** 2022-11-17

**Authors:** Yin Yuan, Wei Wang, Tingting Song, Yongxin Li

**Affiliations:** Institute of Psychology and Behavior, Henan University, Jinming Avenue, Kaifeng 475000, China

**Keywords:** parental burnout, problem behavior, family functioning, psychological aggression, self-control, co-parenting

## Abstract

Parental burnout refers to the feelings of extreme exhaustion that many parents experience due to parenting. Although parental burnout has received considerable interest from researchers, the effects and mechanisms of parental burnout on adolescents’ physical and psychological development remain largely unknown. This study investigated the relationship between parental burnout and adolescents’ problem behaviors in Chinese families. We explored the mediating roles of family function, parental psychological aggression, and adolescent self-control, as well as the moderating role of co-parenting. Questionnaires were distributed to 600 adolescents and their primary caregivers, and the data were collected on three different occasions. In total, 174 paired participants completed the survey (44.9% boys; Mean-age = 14.18 years). Bootstrapping results showed that parental burnout was positively associated with adolescents’ problem behavior. In addition, family function, parental psychological aggression, and adolescent self-control mediated the relationship between parental burnout and adolescents’ problem behavior. Co-parenting moderated the effects of parental burnout on family function and parental psychological aggression. The result of structural equation modeling (SEM) generally supported the above results. These findings reveal the negative effects and mechanisms of parental burnout on adolescent problem behavior, providing empirical evidence that can be applied to prevent the negative effects of parental burnout.

## 1. Introduction

The birth of a child and the accompanying parenting bring not only happiness to parents but also pressure and challenges [1]. When parenting pressure reaches its peak, parents may feel tired and powerless, fail to cope with parenting stress, and develop parental burnout [2]. Parental burnout is defined as a series of symptoms related to parenting, such as exhaustion in their parental role, a low sense of competence and accomplishment, and emotional distance between parents and their children [3]. Ever since Roskam [3] developed a measurement for parental burnout, this topic has received considerable interest from many researchers worldwide [4]. Research on parental burnout mainly focuses on the development of measurements [2,5], the antecedents of parental burnout [6,7], and parental burnout during the coronavirus disease 2019 (COVID-19) pandemic [8]. Prior studies have paid relatively little attention to the consequences of parental burnout, especially its relation to adolescents’ development. However, the healthy development of children is the main objective of parenting, and the negative consequences of parental burnout affect the physical and psychological development of children [9]. Therefore, it is important to explore the direct effect of parental burnout on children’s development. Further, because adolescence is a critical period in individuals’ development, where rapid development occurs, adolescents may face more internal and external contradictions and conflicts, leading to more behavioral and emotional problems [10]. To the best of our knowledge, only two studies have examined the relationship between parental burnout and adolescents’ problem behavior, in which parental burnout influenced adolescents’ problem behavior directly [11], or through maternal hostility [12]. The underlying mechanisms between the two variables need further exploration.

In the family system, family-level variables are relatively macro, variables related to parents are relatively meso, and variables related to adolescents were relatively micro. Therefore, we chose family function, parental psychological aggression, and adolescent self-control as the mediating variables between these three levels. Therefore, the present study examined the relationship between parental burnout and adolescents’ problem behavior, as well as the mediating effects of the family (family function), parents (parental psychological aggression), and children (adolescents’ self-control), and the moderating effect of co-parenting. 

### 1.1. Relation of Parental Burnout and Adolescents’ Problem Behavior

Parental burnout results from a chronic imbalance of risks over resources in the parenting domain [13]. Based on the Balance Between Risks and Resources (BR^2^) theory of parental burnout, parenting stress did not necessarily result in parental burnout when parents had enough resources to cope with parenting stress. Problem behaviors are defined as abnormal behaviors, with severity and duration beyond the age range permitted by social moral norms, which can be divided into internalizing and externalizing problem behaviors [14]. Internalizing problem behaviors indicate behaviors related to anxiety and mood disorders, including depression, anxiety, and withdrawal [15]. Such negative emotions are directed inward, rather than toward others [16]. Externalizing problem behaviors indicate those disruptive behaviors, including aggression and skipping classes [17], where the negative emotions are directed toward others [16].

Previous empirical studies indicated higher parenting stress may be associated with increased adolescent problem behaviors such as social withdrawal, aggression, defiance [18], and decreased prosocial behaviors [19]. This, over time, causes mental health problems such as depression and anxiety in adolescents [20]. Considering that parenting stress is a key antecedent of parental burnout [3,21], the relationship between parental burnout and adolescents’ problem behavior may be stronger than the current observed relationship between parenting stress and adolescents’ adjustment. Prior studies have shown that maternal parental burnout can predict later increases in adolescents’ internalization of problem behaviors [12]. A series of cross-sectional studies also suggested a relationship between parental burnout and problem behaviors. For example, parental burnout can increase adolescents’ anxiety and loneliness, aggressive behavior, and depression levels, thus reducing adolescents’ life satisfaction and mental health levels [9,21,22]. These variables may reflect various aspects of adolescents’ problem behavior. Therefore, we propose: 

**Hypothesis** **1.**
*Parental burnout has a significant positive predictive effect on adolescents’ problem behaviors.*


### 1.2. Mediation Effect of Family Level: The Role of Family Function

Family function is the effectiveness of emotional connections, family rules, family communication, and coping with external events in the family system among family members [23]. Family functions have a great impact on the physical and mental development of adolescents. Good family function means that a family provides stable environmental conditions for the physical and mental health and social development of family members. Good family function can help young people cope with the pressures of school and life, promoting their physical and psychological development [23]. Low family function refers to poor intimate relationships between family members and poor family adaptability, poor family communication and parent–child relationships, and a lack of emotional support from other family members. When facing problems, adolescents in these situations may display more problem behaviors [24].

According to family system theory [25], the negative effects of parental burnout may go beyond the caregivers themselves and affect their relationships with their partners and children; that is, they can affect the whole family system [26]. Prior studies have shown that parental burnout affects several aspects of family function. For instance, parental burnout may increase marital conflict and reduce marital satisfaction [22], leading to neglect and even violent behavior directed at children [26]. The studies cited above suggest that parental burnout undermines the coordination and support of various family systems [27], resulting in a dysfunctional family system. In addition, researchers have also indicated that parental burnout can affect the quality of family function by affecting family communication, organization, and satisfaction [28]. Therefore, we can derive that parental burnout may have a significant negative predictive effect on family function.

Parenting activities are mainly conducted within the family [27]. Family is an important setting for personal growth and socialization and is the most direct and closest micro-environment affecting a person’s psychological development [28]. A low level of family function means a low level of intimacy, mutual support, and cooperation among family members. A dysfunctional family can decrease family members’ care for each other and reduce the self-esteem and self-acceptance ability of adolescents, which may increase problem behavior. In addition, Prior studies have shown that better family function results in positive and harmonious communication, less conflict, higher levels of support among family members, a good environmental basis for adolescents’ psychological growth [29], and fewer adolescent problem behaviors [30]. Thus, we propose: 

**Hypothesis** **2.**
*Parental burnout indirectly affects adolescents’ problem behavior through family function.*


### 1.3. Mediation Effect of Parents’ Level: The Role of Psychological Aggression

Parental psychological aggression refers to the psychological or emotional rejection of children by their parents through verbal (e.g., swearing at child) or symbolic (e.g., threatening to hit a child, but not hitting them) aggression, which is a form of harsh parental discipline [31]. A prior study showed that parental burnout can lead to a series of changes in parenting behaviors and that parenting styles are associated with parental burnout [32]. Furthermore, research has shown that parental burnout increases parental aggression by influencing parents’ experience and expression of anger [22]. Parental burnout also increases parental neglect and violence toward children [26]. Therefore, parental burnout may positively predict parents’ psychological aggression toward their children.

In addition, parents’ parental burnout may affect children’s problem behavior through psychological aggression in several aspects. First, adolescents need to feel their parents’ continuous positive response, their parents’ “acceptance”, in the process of growth and development. However, burned-out parents are unwilling or incapable of engaging in emotional communication and interaction with adolescents, which may increase their psychological aggression toward their children. All of these behaviors give children a feeling of “rejection”, leading them to believe that they are not accepted. This results in a decrease of children’s self-esteem and self-efficacy [33,34], and consequently, in increased displays of problem behaviors. Second, parents who are tired of parenting have reduced sensitivity to their childcare responsibilities and are psychologically aggressive toward their adolescents, leading to situations in which adolescents are not able to establish safe and positive attachment relationship with their parents. This weakens their emotional security and increases the probability of adolescents displaying problem behaviors [35]. Third, adolescents who have experienced parental psychological aggression may learn from their parents and regard aggressive conflict resolution strategies as strategies for problem solving. Such adolescents may regard aggressive conflict resolution strategies as effective problem-solving strategies in their interactions with others, thus increasing their problem behaviors. Prior studies have also shown that parental psychological aggression can significantly and positively predict children’s internalization of problem behavior [36,37,38]. Therefore, parental burnout may increase parents’ psychologically aggressive behavior toward their children, which increases adolescents’ problem behavior. Therefore, we propose: 

**Hypothesis** **3.**
*Parental burnout indirectly affects adolescents’ problem behavior through parents’ psychological aggression toward their children.*


### 1.4. Mediation Effect of Adolescents’ Level: The Role of Self-Control

Self-control refers to individuals’ ability to constrain and manage their behaviors, emotions, and cognitive activities per social standards or their own will [39]. Family is a key proximal influencing factor in individuals’ development of traits [30], and adolescents’ development of self-control is influenced by family factors. Burned-out parents may have a series of behavior changes such as emotional distancing, neglect, and violence toward children, which may be short- or long-term stressful situations for adolescents [3,26]. Because parental burnout results from chronic risk factors overwhelming resource factors, when burned-out parents face stressful situations with their limited psychological resources, they may neglect to parent their adolescents, whose self-control could then decrease [40]. Therefore, parental burnout may negatively affect adolescents’ self-control ability.

Self-control plays a fundamental role in adolescents’ development. A prior study showed that self-control is the primary individual-level cause of crime and that its effect is contingent on criminal opportunity [41]. Research shows that individuals with low self-control are more likely to show emotional and social adaptation problems [42,43] and may also show more deviant, criminal, and other problem behaviors [42,44]. Adolescents’ self-control levels can significantly predict adolescents’ problem behaviors. Therefore, it can be speculated that parental burnout leads to the reduction of adolescents’ self-control, which then increases the possibility of adolescents’ adopting and displaying problem behaviors. Therefore, we propose: 

**Hypothesis** **4.**
*Parental burnout indirectly affects adolescents’ problem behavior by reducing adolescents’ self-control ability.*


### 1.5. The Moderation Effect of Co-Parenting

Co-parenting refers to the degree of coordination and cooperation between parents in parenting objectives, concepts, attitudes, and methods in the process of raising children [45]. Although in a family, parenting activities related to children may mainly be completed by either the father or mother, this does not mean that the other parent is not involved in parenting activities. On the contrary, raising children is the common responsibility and obligation of both parents, spouses, or other family members, as primary caregivers are important sources of social support, and spouses’ helping behaviors may play an important supporting role in parenting [46]. 

Co-parenting is an executive subsystem that regulates interaction within the family system [47]. The ecological model of co-parenting indicates that the quality of co-parenting plays an important role in parental mental health [48]. High levels of co-parenting can reduce parenting stress levels [49]. Research has shown that when parenting goals and practices between parents are consistent, caregivers’ parenting stress decreases [50]. When one parent suffers high levels of parental burnout, it may cause dysfunction in the whole family system, increase the possibility of parents being psychologically aggressive toward their children, affecting their children’s self-control, and thus, threatening children’s healthy development. However, when a primary caregiver’s spouse provides effective support, it may buffer the negative impact of parental burnout. Thus, we propose: 

**Hypothesis** **5.**
*Co-parenting moderates the relationship between parental burnout and parents’ psychological aggression, adolescents’ self-control, and adolescents’ problem behavior.*


### 1.6. The Present Study

To sum up, based on family systems theory and defining a family’s primary caregiver, this study collected the paired data of “primary caregivers” and “adolescents” at three different time points. Because adolescents’ psychological maturity does not match their physical maturity [51], they may develop a series of behavior problems, thus increasing the parental pressure on their parents. Therefore, adolescents’ parents are a suitable sample for research on parental burnout. This study investigates the effects and underlying mechanisms of parental burnout on adolescents’ problem behavior. Specifically explored are the mediating roles of family function, parental psychological aggression, and adolescents’ self-control, as well as the moderating role of primary caregivers’ spouse’s co-parenting. Figure 1 illustrates the research framework.

## 2. Materials and Methods

### 2.1. Participants and Procedures

Cluster sampling was used to recruit the sample, which consisted of junior middle school students, and their primary caregivers (father or mother), from a middle school in Henan Province, China. To avoid common method bias, data were collected at three different time points, with an interval of half a month between each point. At Time 1, data on parents’ parental burnout and co-parenting (answered by parents) were collected. A total of 600 questionnaires were distributed, and 546 questionnaires were recovered. At Time 2, data on family function, parents’ psychological aggression (answered by parents), and self-control (answered by adolescents) were collected. A total of 600 questionnaires were distributed and 526 questionnaires were recovered. At Time 3, data on adolescents’ problem behaviors (answered by adolescents) were collected. A total of 600 questionnaires were distributed, and 519 questionnaires were recovered.

All questionnaires with incomplete answers or missing data were eliminated, resulting in 493 groups of valid questionnaires being retained. In addition, to reduce the impact of extreme values on the results, we removed data with scores greater than three times standard deviations on each item, and 41 groups of questionnaires were removed. 

The questionnaires of both students and parents contained the item “primary caregiver”. For the first survey, students were asked to take the questionnaires home and hand them to their primary caregivers. Parents were asked to indicate the child’s primary caregivers, and they had the options “themselves”, “spouse”, or “other”. For the third survey, students were also asked who their primary caregivers were. A total of 78 responses regarding the identification of the primary caregiver did not correspond from parent to children. These were excluded. In total, 374 pairs of data were retained. Among the primary caregivers, 316 were mothers and 58 were fathers. Regarding the adolescents, there were 168 boys and 206 girls aged 14.18 ± 0.60 years.

The survey was approved by the Research Ethics Committee of the academic institution that the authors are affiliated with. Informed consent was obtained from all participants.

### 2.2. Parental Burnout

Parental burnout was measured using the Chinese short version of the Parental Burnout Assessment (s-PBA, [52]). It has one factor, consisting of seven items, and each item was rated using a seven-point Likert scale ranging from 1 (completely inconsistent) to 7 (completely consistent), the average score was calculated, and a higher score represented higher burnout. An example item is, “When I woke up in the morning and thought of taking care of my children all day again, I felt very tired”. Cronbach’s α for the measure was 0.77 in the current study.

### 2.3. Problem Behavior

Adolescents completed the Chinese version [53] of the Youth Self Report [54]. The original questionnaire consists of eight factors with 112 items. The present study measures two dimensions: internalized problem behavior (31 items; e.g., “I often cry”) and externalized problem behavior (22 items; e.g., “I often argue”). All items were scored on a 3-point Likert scale, with 0 being “never” and 2 being “often”. The average score was calculated, and higher scores represented more serious adolescent problem behaviors. For the current sample, Cronbach’s α for the measure was 0.89.

### 2.4. Family Function

The second edition of the Family Cohesion and Adaptability Scale (FACES II) was compiled by Olson [55]. The revised FACES II has good reliability and validity [56]. The scale includes 30 items (e.g., “in our family, everyone participates in entertainment activities”). The items were scored on a 5-point Likert scale, where 1 = “no”, and 5 = “always”. The average score was calculated, and higher scores represented better family adaptability and cohesion. In this study, Cronbach’s alpha of the scale was 0.91.

### 2.5. Psychological Aggression

The Parent–Child Conflict Tactics Scale [57] was used to measure parents’ harsh disciplinary behaviors. The 22–item CTSPC contains five subscales: psychological aggression, corporal punishment, nonviolent discipline, severe physical assault, and severe physical assault, of which the Psychological Aggression subscale (5 items, e.g., “shouting, yelling, or screaming”) was the main focus of the present study. Parents reported how often they employed psychologically aggressive behaviors toward their children in the past year. The items were scored on a 7-point scale, as follows: never (0); once (1); twice (2); 3–5 times (4); 6–10 times (8); 11–20 times (15); and  > 20 times (25). The frequencies were calculated by summing the scores of the subscale items. In this study, Cronbach’s alpha of the scale was 0.77.

### 2.6. Self-Control

Adolescents’ self-control was assessed using the Self-Control Scale [44]. All participants answered the Chinese version of the SCS, which has been used in Chinese adolescent populations and has shown good reliability and validity [58]. It consists of 19 items (e.g., I can work long-term for one goal), rated on a 5-point Likert scale (from “1  =  not like me at all” to “5  =  like me very much”). The average score was calculated, and a higher total score represented stronger adolescent’s self-control. Cronbach’s α for this scale in this study was 0.86.

### 2.7. Co-Parenting

Parents completed the Chinese version [46] of the Co-parenting Relationship Scale [59], which measures the quality of parents’ co-parenting experiences. The scale includes 14 items (e.g., “my spouse cares about our children very much”) on a 7-point Likert scale (0  = “completely inconsistent” through 6  = “completely consistent”). The original scale was filled in by mothers evaluating fathers’ daily co-parenting performances. Since, for this study, the individual who completed the scale could be the father, item expressions were altered accordingly. The average score was calculated, and a higher score indicated a more supportive co-parenting experience. Cronbach’s α was 0.80.

### 2.8. Data Analysis

All data analyses were conducted using IBM SPSS 23.0 and AMOS 23.0. First, common method bias was analyzed using Harman single-factor test, then, descriptive statistics and correlation analysis were performed. Second, PROCESS macro was used to test the mediation effects and the moderated mediation effects, respectively. Finally, AMOS 24.0 was used to establish a structural equation model for parental burnout, adolescent problem behavior, family function, parental psychological aggression, adolescent self-control, and co-parenting. The Bootstrap method was used to test the significance of different pathways.

## 3. Results

### 3.1. Common Method Bias

Although this study used paired data from different sources and a multiple time point method to collect data, all questionnaires were self-assessment scales that could produce common method bias. Therefore, common method bias was examined using the Harman single-factor test [60]. The results of the exploratory factor analysis showed that the number of factors without rotation was greater than 1, and the variance interpretation percentage of the first principal component was 14.83%, less than 40% [55]. This indicates that common method bias had little effect on the overall results of the present study.

### 3.2. Descriptive Statistics and Correlation Analysis

The descriptive statistics and correlation matrix of each variable are shown in Table 1. The results showed that parental burnout was negatively correlated with family function, self-control, and co-parenting, and positively correlated with parental psychological aggression and adolescents’ problem behavior. Adolescents’ problem behavior was negatively correlated with family function and self-control, and positively correlated with parental psychological aggression. These results preliminarily supported our hypotheses.

### 3.3. Hypotheses Testing for Mediating Effects

Linear regression analysis was conducted to examine the relationship between parental burnout and adolescents’ problem behavior. The results showed that parental burnout was positively associated with adolescents’ problem behavior (*β* = 0.23, *t* = 4.48, *p* < 0.001), supporting Hypothesis 1. To examine the hypothesized model, Model 4 of the PROCESS macro [61] was used. Parental burnout was the independent variable and family function, parental psychological aggression and adolescents’ self-control were the mediating variables, with adolescents’ problem behavior as the dependent variable. Following the suggestions of Fang et al. [62], a non-parametric bootstrapping method (*n* = 5000) was used with a 95% confidence interval calculated using the bias-corrected bootstrapping method. 

As the results show in Figure 2, Figure 3 and Figure 4, family function could mediate the relationship between parental burnout and adolescents’ problem behavior (mediation effect = 0.03, SE = 0.01, 95% CI [0.008, 0.062]). Parents’ psychological aggression could mediate the relationship between parental burnout and problem behavior (mediation effect = 0.04, SE = 0.01, 95% CI [0.018, 0.075]. Self-control could mediate the relationship between parental burnout and adolescents’ problem behavior (mediation effect = 0.11, SE = 0.02, 95% CI [0.066, 0.158]).

### 3.4. Hypotheses Testing for the Moderated Mediating Effect

Based on the above analysis of the mediating effects, Model 7 of PROCESS macro was used to test the moderated mediating effect. The results are presented in Table 2. The interaction between parental burnout and co-parenting was significantly associated with family function (*β* = −0.12, *t* = −2.21, *p* < 0.05), suggesting that co-parenting could moderate the relationship between parental burnout through family function. The interaction between parental burnout and co-parenting was also significantly associated with parental psychological aggression (*β* = 0.14, *t* = 2.34, *p* < 0.05), indicating that co-parenting could moderate the relationship between parental burnout and parental psychological aggression through family function. However, the interaction between parental burnout and co-parenting did not significantly predict adolescents’ self-control (*β* = −0.07, *t* = −1.22, n.s.).

Simple slope tests were conducted to further explore the moderating effect of co-parenting (Table 3). We investigated the impact of parental burnout on family function at different co-parenting levels. The specific moderating effects are shown in Figure 5. For primary caregivers receiving high levels of co-parenting, parental burnout was significantly associated with family function (*β* = −0.28, *p* < 0.01, 95% CI [−0.48, −0.09]). With low levels of co-parenting, parental burnout was not significantly associated with family function (*β* = −0.02, n.s., 95% CI [−0.15, 0.10]). This result suggested that co-parenting could buffer the negative effect of parental burnout on family function.

Figure 6 shows the moderating effect of co-parenting on the relationship between parental burnout and psychological aggression. With high level of co-parenting, parental burnout was significant associated with psychological aggression (*β* = 0.36, *p* < 0.001, 95% CI [0.16, 0.56]). With low co-parenting levels, parental burnout was not associated with psychological aggression significantly (*β* = 0.06, n.s., 95% CI [−0.07, 0.20]). This result suggests that co-parenting could buffer the negative effect of parental burnout on psychological aggression.

### 3.5. Hypotheses Testing of Structural Equation Model (SEM)

The SEM was conducted to further test the effects and mechanisms of primary caregivers’ parental burnout on adolescents’ problem behavior. Parental burnout was set as an independent variable. Family function, parental psychological aggression, and adolescents’ self-control were set as mediating variables, with adolescents’ problem behavior being set as the dependent variable. The interaction between parental burnout and co-parenting was added and regressed to the mediation variables.

The model showed acceptable goodness of fit (*χ*^2^ = 4.34, *df* = 5, n.s., CFI = 1.00, RMSEA = 0.00). The results of the path analysis are presented in Figure 7. Parental burnout was significantly associated with family function (*β* = −0.19, *p* < 0.01), parental psychological aggression (*β* = 0.21, *p* < 0.01), and adolescents’ self-control (*β* = −0.25, *p* < 0.001). Parental psychological aggression and adolescents’ self-control (*β* = 0.33, *p* < 0.001) were positively associated with problem behavior (*β* = 0.25, *p* < 0.001). Meanwhile, the interaction between parental burnout and co-parenting was significantly associated with family function (*β* = −0.28, *p* < 0.05), parental psychological aggression (*β* = 0.24, *p* < 0.05), and adolescents’ self-control (*β* = −0.23, *p* < 0.05). Bootstrapping analyses (*n* = 5000) were conducted to examine the mediation effects, and the 95% CI values were calculated using the bias-corrected bootstrapping method. The results showed that parental psychological aggression and adolescents’ self-control could mediate the relationship between parental burnout and problem behavior (Table 4).

## 4. Discussion

Children are the subjects of parenting behavior, and burned-out parents’ parenting behavior ultimately affects the healthy development of their children. However, only a few studies have examined the direct effects of parental burnout on children’s development [12]. Therefore, based on family system theory, this study used paired data to build a moderated mediation model to explore the influence and underlying mechanism of parental burnout on adolescents’ problem behavior. The results generally supported the hypotheses and showed that parental burnout may affect adolescents’ problem behavior, both directly and indirectly, through family function, parents’ psychological aggression, and adolescents’ self-control. Furthermore, the mediating effects of parental psychological aggression and adolescents’ self-control are moderated by co-parenting.

### 4.1. The Mediation Effect of Family Function

Burned out parents are prone to negative emotions, such as anxiety, and tend to escape from parenting situations [3,26], resulting in reduced intimacy and warm expressions toward family members (both their spouse and children) [3,17,21,63,64]. Burned out parents’ lack of emotional involvement in their family harms the degree of emotional connection between family members, breaking a stable family structure, affecting the family environment and atmosphere, and lowering the family function of the whole family [28]. 

Family is an important environment for individual growth and socialization [65]. However, the result of the structural equation model suggests that the predictive effect of family function on adolescent problem behavior is insignificant. It can be speculated that the variance of problem behavior explained by family function is shared with parental psychological aggression and adolescent self-control. In other words, the effect of family function on adolescent problem behavior may be affected by two other mediating pathways. 

Adolescent development is the result of continuous interaction with the surrounding environment. As noted above, an ecosystem can be divided into a distal environment and a proximal environment according to the environment’s closeness of contact with the individual. Here, in the family, the family environment and family function could be viewed as the distal environment, and the parent–child interaction could be viewed as the proximal environment. Distal factors in an ecosystem influence individual development through the proximal environment [66]. As a distal factor measuring the functioning of a family ecosystem [67], family function may affect the personal growth of adolescents through parental psychological aggression and adolescents’ self-control (proximal factors). Future studies could further explore the relations between the mediation variables in this study. 

### 4.2. The Mediation Effect of Parental Psychological Aggression

This study found that parental psychological aggression plays a mediating role between parental burnout and adolescents’ problem behavior. This finding is consistent with previous studies that found that parental burnout leads to a range of negative parenting behaviors, such as reduced warmth and support for children [3], increased neglect and violence [4], hostility [12], aggression [18], and increased negative treatment of children, such as denial and rejection [4]. This study shows that parental burnout can also increase parents’ psychological aggression toward adolescents. Such negative parenting behaviors may be due to parents’ poor coping styles in the face of parenting exhaustion [12].

### 4.3. The Mediation Effect of Self-Control

The results show that adolescents’ self-control plays a mediating role between parental burnout and adolescents’ problem behaviors. Specifically, high levels of parental burnout result in reduced self-control in adolescents, which leads to adolescents displaying more problem behaviors. According to the BR^2^ theory, burned-out parents strive to restore the balance of resources and risk. They may try to escape from parenting, reducing their emotional connection with their children and resulting in parents rejecting and neglecting their children, and sometimes even displaying violent behavior toward their children [3,26]. Such parents also tend to adopt negative parenting styles. All of the above parental behaviors can be long-term pressures that adolescents have to face, which could continuously occupy and consume adolescents’ psychological resources, affecting their development of self-control. Low self-control can further trigger adolescents’ problem behavior. Such pressures are inconsistent with the children’s needs. When adolescents perceive such rejection, they experience a series of negative emotions, such as anger, helplessness, and anxiety, and these negative emotions lead to a reduction of adolescents’ sense of emotional security [68]. Due to limited self-control resources, adolescents are unable to deal with challenges well, making them prone to externalizing problem behaviors, such as aggression, and/or internalizing problem behaviors, such as depression, when facing provocative or scary situations [69]. Burned-out parents with limited resources may fail to regulate their own behaviors. Adolescents acquire these cognitive and behavioral patterns, which are manifested as low self-control, leading to greater internalization and externalization of problem behaviors. Therefore, future research should inform the strengthening of interventions directed at parental burnout to both prevent and mitigate its effects.

### 4.4. The Moderation Effect of Co-Parenting

Co-parenting was selected as a moderating variable in this study because it can play a positive role in parenting activities [46]. For example, co-parenting from grandparents can effectively reduce the depression levels of caregivers [70]. In a study on mothers, father involvement in parenting could not only provide mothers with necessary familial support, but also effectively reduce the mothers’ parenting stress levels. In addition, father involvement in parenting contributes to the physical and mental development of children [12]. The empirical results of this study partially support the moderating effect of co-parenting. Co-parenting moderates the effects of parental burnout on family function and psychological aggression.

Based on the mediating model of family function, co-parenting moderates the effect of parental burnout on family function. First of all, in cases of both low and high parental burnout, high co-parenting resulted in higher family function than low co-parenting. That is, co-parenting could buffer the negative effects of parental burnout on family function. Specifically, the moderating effect of high co-parenting conforms to the idea of “a drop in the bucket”. In other words, the protective effect of high co-parenting only plays a significant role in low parental burnout and gradually weakens as parental burnout gradually increases. When parental burnout reaches a certain level, the spillover effect of parental burnout on the family system cannot be weakened even if others cooperate in parenting. The results of this study suggest that we should improve co-parenting levels so that parents may reduce the negative effects of parental burnout. This would improve both family function and the physical and mental health of adolescents.

The results also showed that co-parenting had a significant moderating effect between parental burnout and parents’ psychological aggression. We initially expected that co-parenting would buffer the negative effects of parental burnout; that is, with a high level of co-parenting, psychological aggression would not increase despite an increase in parental burnout. However, the moderation effect was not per our expectation. The results showed that when co-parenting was low, psychological aggression did not change with increasing parental burnout. However, when co-parenting was high, psychological aggression increased with increasing parental burnout. This may be because a high level of co-parenting represents the high-level of coordination and cooperation between father and mother, and they showed stronger consistency than those with low levels of co-parenting. When the primary caregiver’s parental burnout was high, their parental burnout would transmit to their partner, and they just wanted to withdraw their affection and escape from their parenting responsibilities [3]. This may increase their level of psychological aggression toward their children. However, future research is needed to examine whether this assumption can be supported. 

This indicates that the resources provided by co-parenting do not buffer the negative effects of parental burnout on adolescent self-control, suggesting that co-parenting may have a more significant effect on the parents and family as a whole, but less of an effect for adolescents. Therefore, future research could explore factors (e.g., adolescents’ emotional regulation efficacy and temperament type) that more closely related to children’s self-control to explore this moderating effect.

### 4.5. Theoretical and Practical Implication

The present research results have important theoretical and practical implications. This research provides important empirical evidence for how the manifestations of parental burnout affect adolescents’ problem behavior. This could be used to inform future practice and parent-child interventions. First, at the family level, family members should try their best to support each other and create an atmosphere in which they can express their opinions to improve family functions. Second, parents should improve their coordination and consistency by consulting and supporting each other, to improve their level of co-parenting. Family therapists could also focus on improving families’ co-parenting levels to help burned-out parents and improve family function. Third, we should prevent the negative spillover effect of parental burnout on each family subsystem and design interventions addressing the three levels of family, parents, and children. As high levels of co-parenting buffer the negative effects of parental burnout, parents should be encouraged to improve their co-parenting and provide parenting support to each other. Parents should also enhance their understanding and pay attention to the psychological state of their spouse to prevent the occurrence of excessive parental burnout.

### 4.6. Limitations and Future Directions 

This study informs our understanding of the influencing mechanisms of parental burnout on adolescents’ problem behavior. Still, the study has some limitations. First, although a multi-source, multi-time data collection method was used in this study to reduce common method bias, the data were all self-reported. Future research may consider using data collection methods combining self-rating and other scales to further explore this topic. Second, this study selected adolescents and their primary caregivers as the research participants because parental rearing in adolescence puts forward higher requirements on parents in many aspects, making it ideal for research on parental burnout [20]. However, it should be noted that the occurrence of parental burnout is not limited to adolescence and future research could select children in different age groups and their parents to broaden and deepen discussions on the relationship between parental burnout and children’s growth. Third, although this study makes use of a multi-time data collection method, which satisfies the basic requirement of causal inference, causal inference is difficult to draw and it is difficult to rule out other possible paths. Whether there are any other intermediary and regulating variables remains to be explored. A longitudinal follow-up study could further improve the robustness of research on this topic.

## 5. Conclusions

In conclusion, this study makes an important contribution to the exploration of the relationship between parental burnout and adolescent problem behavior. Specifically, the results of this study show that parental burnout can affect adolescents’ problem behaviors through three dimensions: family, parents, and children. This study enriches our understanding of the variable consequences of parental burnout, especially in adolescents. More research is needed to enrich our understanding of the impact of parental burnout on children’s physical and mental development in order to aid in the development of resources and intervention strategies that can reduce adverse outcomes.

## Figures and Tables

**Figure 1 ijerph-19-15139-f001:**
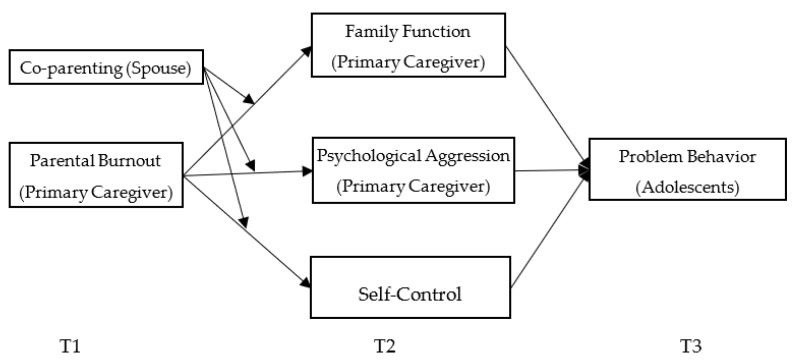
The basic framework of this study.

**Figure 2 ijerph-19-15139-f002:**
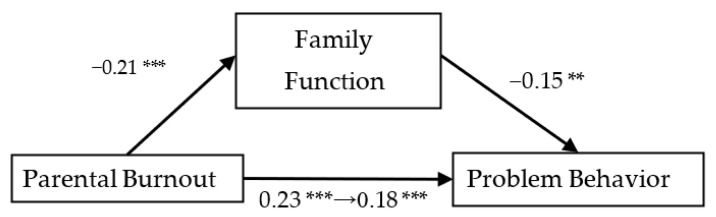
The mediating effect of family function on parental burnout and problem behavior. ** *p* < 0.01, *** *p* < 0.001.

**Figure 3 ijerph-19-15139-f003:**
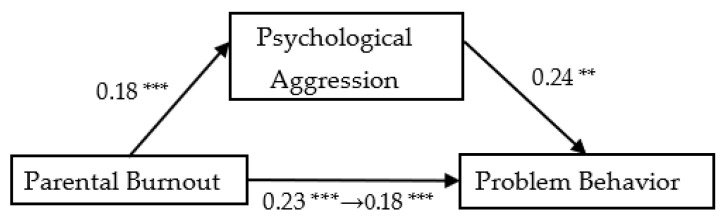
The mediating effect of psychological aggression on parental burnout and problem behavior. ** *p* < 0.01, *** *p* < 0.001.

**Figure 4 ijerph-19-15139-f004:**
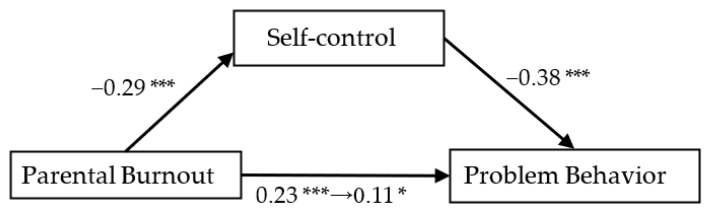
The mediating effect of self-control on parental burnout and problem behavior. * *p* < 0.05, *** *p* < 0.001.

**Figure 5 ijerph-19-15139-f005:**
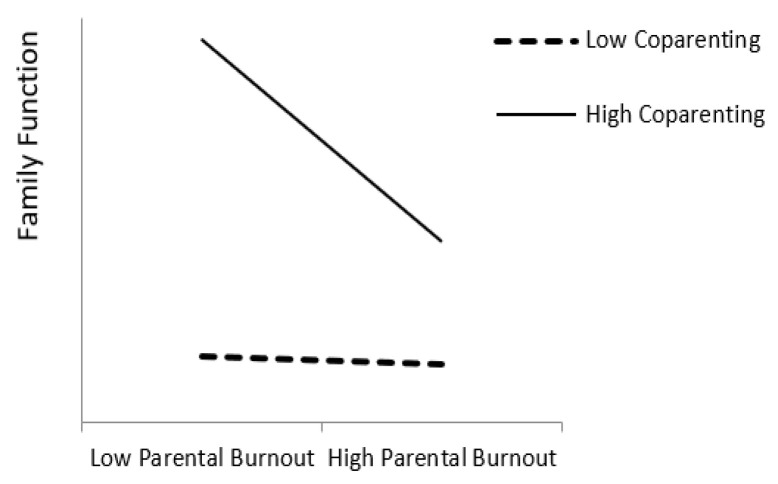
The moderating effect of co-parenting on parental burnout and family function.

**Figure 6 ijerph-19-15139-f006:**
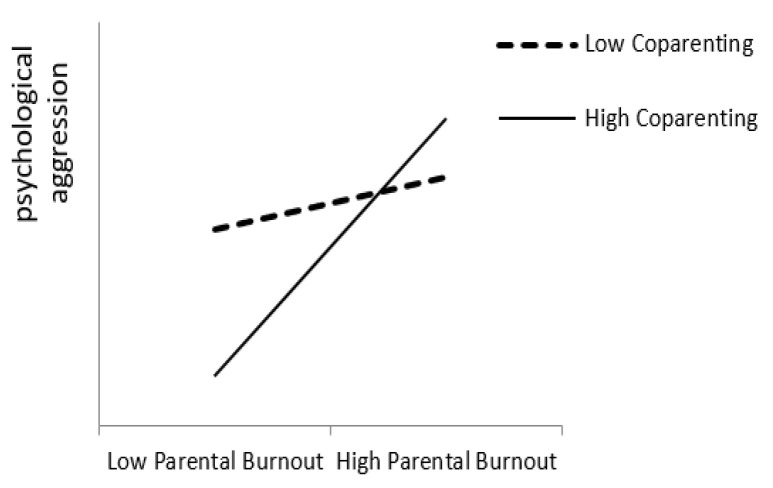
The moderating effect of co-parenting on parental burnout and psychological aggression.

**Figure 7 ijerph-19-15139-f007:**
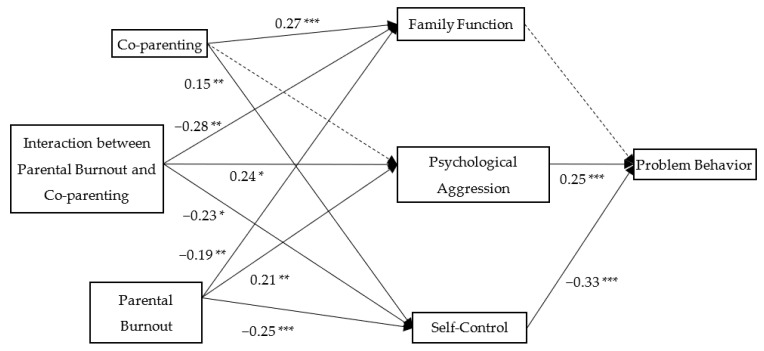
Standardization path model. * *p* < 0.05, ** *p* < 0.01, *** *p* < 0.001.

**Table 1 ijerph-19-15139-t001:** Descriptive statistics and correlations between study variables.

		M ± SD	1	2	3	4	5	6
1	Parental Burnout	1.47 ± 0.59	1					
2	Co-parenting	4.46 ± 0.93	−0.34 **	1				
3	Family Function	5.00 ± 0.62	−0.21 **	0.36 **	1			
4	Parental Psychological Aggression	2.42 ± 3.60	0.16 **	−0.15 **	−0.27 **			
5	Adolescents’ Self-Control	3.62 ± 0.54	−0.26 **	0.27 **	0.32 **	−0.19 **	1	
6	Adolescents’ Problem Behavior	0.33 ± 0.18	0.21 **	−0.16 **	−0.19 **	0.27 **	−0.41 **	1

** *p* < 0.01.

**Table 2 ijerph-19-15139-t002:** Testing the moderated mediation effects of parental burnout on problem behavior.

Regression Equation		Fitting Index			Significance of Regression Coefficient		
Dependent Variable	Independent Variable	R	R^2^	F	*β*	95% CI	*t*
Family Function	Parental Burnout	0.40	0.16	19.97 ***	−0.15 *	[−0.27, −0.04]	−2.58
	Co-parenting				0.31 ***	[0.20, 0.42]	5.74
	Parental Burnout × Co-parenting				−0.12 *	[−0.23, −0.01]	−2.21
Adolescents’ Problem Behavior	Parental Burnout	0.27	0.07	12.30 ***	0.18 ***	[0.08, 0.29]	3.42
	Family Function				−0.15 **	[−0.25, −0.04]	−2.79
Parental Psychological Aggression	Parental Burnout	0.23	0.05	6.30 ***	0.21 **	[0.09, 0.33]	3.45
	Co-parenting				−0.06	[−0.17, 0.05]	−1.02
	Parental Burnout × Co-parenting				0.14 *	[0.02, 0.26]	2.34
Adolescents’ Problem Behavior	Parental Burnout	0.32	0.10	19.56 ***	0.18 ***	[0.08, 0.28]	3.45
	Parental Psychological Aggression				0.24 ***	[0.13, 0.34]	4.51
Adolescents’ Self-Control	Parental Burnout	0.34	0.12	14.98 ***	−0.25 ***	[−0.37, −0.14]	−4.30
	Co-parenting				0.18 **	[0.07, 0.29]	3.24
	Parental Burnout × Co-parenting				−0.07	[−0.18, 0.04]	−1.22
Adolescents’ Problem Behavior	Parental Burnout	0.43	0.19	40.01 ***	0.11 *	[0.01, 0.21]	2.25
	Adolescents’ Self-Control				−0.38 ***	[−0.48, −0.28]	−7.65

* *p* < 0.05, ** *p* < 0.01, *** *p* < 0.001.

**Table 3 ijerph-19-15139-t003:** The Moderating Effect of Co-parenting.

	Co-Parenting	*β*	SE	BootLLCI	BootULCI
Family Function	M − 1SD	−0.02	0.06	−0.15	0.11
	M + 1SD	−0.28	0.10	−0.48	−0.09
Parental Psychological Aggression	M − 1SD	0.06	0.07	−0.07	0.20
	M + 1SD	0.36	0.10	0.16	0.56

**Table 4 ijerph-19-15139-t004:** Detailed results of the bootstrapping method.

Paths	Estimate	95% CI		
		LLCI	ULCI	*p*
Parental Burnout → Parental Psychological Aggression → Adolescents’ Problem Behavior	0.052	0.021	0.096	0.000
Parental Burnout → Adolescents’ Self-Control → Adolescents’ Problem Behavior	0.084	0.041	0.142	0.000

## Data Availability

The datasets generated for this study are available on request to the corresponding author.

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
