# Peer review of "The Mechanisms of Parental Burnout Affecting Adolescents’ Problem Behavior"

_ijerph, 2022, doi:10.3390/ijerph192215139_

Round 1
Reviewer 1 Report
Thank you for the opportunity to review this paper, which I found very interesting to read. This is a well-designed and rigorous study of the relationships between parental burnout and a number of other variables related to family functioning and adolescent behaviour. The paper is largely well-written and structured, with hypotheses and results clearly explained and presented. The study findings are of significant interest and contribute to the empirical evidence base on the consequences of parental burnout, and possible moderating influences. The authors also present some helpful ideas for further research in this area to build on their findings.
This paper has the potential to be a valuable inclusion in the special issue on parental attachment and adolescent wellbeing. I don’t believe the paper is quite ready for publication in its current form but it could be improved relatively easily. Some suggestions for revisions are provide below.
General issues
1. There are a lot of different theoretical frameworks mentioned at various points in the paper with little indication of the relationship, if any, between them. This makes the theoretical framing feel somewhat fragmented. The theoretical strands could be drawn together more to form a more coherent whole, or some of the theoretical references which add little to the interpretation of the findings could be safely omitted.
2. At line 45, the authors say they found only two studies examining the relation between parental burnout and adolescents’ problem behaviour. Was a systematic review conducted? If not, it sounds a bit misleading, as though there are only two such studies to be found. And in fact, at line 65, the authors refer to ‘A series of cross-sectional studies also suggested a relationship between parental burnout and problem behaviors’, which seems to contradict line 45.
3. I would have liked to see a little more framing around the drivers of parental burnout, and a little more definitional clarity. In particular, is a low level of co-parenting/support a cause of parental burnout as well as a mediating influence on its consequences? At line 54, the authors state ‘Parental burnout results from chronic and unalleviated parenting stress’ but surely there are other possible causes or contributing factors? What about parental mental or physical health or disability, or stress from outside the family context? It is not clear how parental burnout is different from chronic parental stress. And does parental burnout refer to people who happen to be parents burning out or burnout arising specifically from parenting stress?
4. In discussion of the self-control variable, the authors sometimes appear to be talking about parental self-control (e.g. lines 127, 469, 484) when the variable is actually adolescents’ self-control.
5. The discussion section is not quite as well structured as the rest of the paper. Some parts (e.g. lines 407-15, most of para starting line 422, lines 444-461) may be better placed in the literature review section where the hypotheses are outlined.
6. There is an allusion at line 525 that implies high levels of co-parenting may sometimes, counter-intuitively, cause more rather than less stress for the primary caregiver. This is confusing as currently framed, particularly because the definition of ‘co-parenting’ is not quite clear. Does ‘co-parenting’ encompass any form of involvement in parenting, not just behaviours that are supportive of the primary caregiver? Does the tool used to measure levels of co-parenting capture only supportive behaviours and attitudes or is it broader? And does the tool also capture co-parenting by people other than the secondary caregiver? E.g. co-parenting by grandparents is mentioned at line 493. I would like to see a clearer definition of co-parenting earlier on (i.e. at line 145).
7. The implications of the findings could be addressed better. At the moment this feel like a bit of an afterthought. The authors highlight the potential benefits of co-parenting quite nicely (e.g. at lines 510, 547), but more insight could be offered into how to promote co-parenting. At lines 540-542, the authors say that their findings could be used ‘to inform future practice and parent-child interventions’ but they say little about how this might work. The line at 544 suggesting that the findings highlight the need for interventions to address all three levels of the family system illustrates a potentially useful insight for practice – a little more commentary like this would add to the paper’s relevance.
Improving definitional and conceptual clarity
Some of the key concepts in the paper could be more clearly defined, explained and differentiated in the literature review section. Examples are:
· ‘Co-parenting’ (as above).
· ‘Problem behaviour’. Why are certain behaviours considered problematic among adolescents? Who decides what is problematic? There may be cultural differences that affect this judgement. And the distinction between internalised and externalised problem behaviour should be clarified.
· How parental burnout differs from very high levels of parental stress, and the difference between adolescent ‘problem behaviour’, ‘maladaptation’ and ‘maladjustment’. There is a bit of slippage between these terms, but sometimes it appears there are material differences in play. E.g. paragraph starting line 53 seems to say there is prior work on the relationship between parental stress and adolescent adaptation, but not on the relationship between parental burnout and adolescent problem behaviour.
Methods
Please include a brief statement of why multiple time points were used in the methods section.
Line 559 – ‘because parental rearing in adolescence puts forward higher requirements on parents in many aspects, making it ideal for research on parental burnout’. I would like to see the reasons for focusing on adolescents unpacked a little more in the methods section rather than at the end of the paper. The particular requirements of parenting adolescents as compared to other age groups could be clarified, with a reference or two, if this is the justification for focusing on this age group.
Some areas where written expression could be improved
Line 34 – ‘from over 100 researchers across 40 countries’. This sounds a little odd. Unless a systematic review was conducted, it is a bit too precise. Suggest rephrasing.
Line 80 – ‘parental burnout destroys the coordination and support of various family systems’. Is destroys a little strong? Perhaps undermines?
Line 86 – ‘Family is the main setting for personal growth and socialization’. Suggest reframing as ‘family is an important setting’. It could be debated whether it is the main setting as adolescents grow older – peers and school have significant effects, for example. There could also be differences across different cultural settings.
Line 90 – ‘higher levels of love and support among family members’. The implication that you can measure love would be better avoided; maybe just leave it at levels of support.
Line 123 – ‘Adolescents’ development of self-control cannot be detached from the influence of family factors’. Suggest reframing as ‘development of self-control is influenced by family factors’. There are other influences outside the family context.
Line 150 – ‘husband and wife’. I suggest using language throughout the paper that avoids the assumption two parents are married, or that they are necessarily a male-female dyad.
Line 158 – ‘spouse and caregiver’. I suggest using language that does not imply one parent takes sole responsibility for caregiving. At other points ‘primary caregiver’ is used and this is preferable.
Line 359 – ‘the negative effect of parental burnout on family function’. Should this be ‘on psychological aggression’?
Line 389 – ‘only a few studies have examined the direct effects of parental burnout on children’s health’. The reference to health is a little confusing because the study relates to adolescent behaviour, not their health.
Line 488 – ‘future research should strengthen interventions directed at parental burnout to both prevent and mitigate its effects’. Suggest rephrasing ‘future research should inform the strengthening of interventions…’
Line 513 – ‘The moderating effect of co-parenting also showed that co-parenting had a significant moderating effect’. Suggest rephrasing ‘The results also showed…’
Line 539 – ‘This research provides important empirical evidence for the mechanisms of parental burnout and how they affect adolescents’ problem behavior’. Suggest rephrasing to avoid implying that the research investigates how parental burnout occurs. Perhaps ‘for how the manifestations of parental burnout affect…’?
Line 575- ‘This study enriches the variable consequences of parental burnout, especially in adolescents’. Suggest rephrasing to ‘enriches our understanding of the variable consequences of parental burnout, especially for adolescents’.
Some minor typographical errors to attend to
Line 43 – ‘adolescences’ should be ‘adolescents’.
Line 113 – should be ‘children’s internalization of problem behavior’.
Line 117 – should be ‘parents’ psychological aggression toward their children’.
Line 160 – should be ‘psychologically aggressive’. There are a few other instances in the paper of this.
Line 349 – ‘caregiver’s’, does not need apostrophe.
In Figure 7, the psychological box should presumably be labelled ‘psychological aggression’.
Line 425 – ‘In the family’, the in should not be capitalised.
Line 454 – ‘leading to adolescents’ not establishing’, no need for apostrophe.
Line 475 – ‘affect’ should be ‘affecting’.
Reviewer 2 Report
The study investigated family function, parental psychological aggression and adolescent self-control as three mediators in the relationship between parental burnout and adolescents’problem behavior by using a sample of adolescents and their primary care-givers. Overall, the findings support the hypotheses. The manuscript was clear and well organized. The authors are to be commended for expanding our knowledge in this important area.
Changes to Consider:
1、In the Introduction section, page 2, the authors mentioned that “We found only two studies examining the relation between parental burnout and adolescents’ problem behavior " however, the present study did not interpret the underlying mechanisms between the two variables.
2、Please explain why choose the family function, parental psychological aggression and adolescent self-controlas three mediators as mediators.
3、Please explain the relationship among parental burnout , adolescents’problem and family function deeply.
4、Please add the detail of measure tool in the manuscript.
5、 please polish language of this paper.
Reviewer 3 Report
I would like to thank the editor for providing me a chance to review the article manuscript.
After carefully reading through the manuscript, I want to express that the study is was well-done and well-written.
I can only point only one minor issue.Some references should be checked carefully.
eg. Line 694 , 49. Wang, W.; Wang, S.; Cheng, H.; Wang, Y.; Li, Y. Short Version Of The Parental Burnout Assessment. Chin ment health J. 2021b. 694 Validation Of The Chinese, 35, 941–946. Line 701,52. Olson, D. H. L. 1985. Family inventories [inventories used in a national survey of families across the family life cycle]. St. Paul, Minn. Family Social Science; University of Minnesota.
Good luck with finalizing the document.
